# Probabilistic Distillation Transformer: Modelling Uncertainties for Visual Abductive Reasoning

Submission Id: 1929*

## ABSTRACT

Visual abduction reasoning aims to find the most plausible explanation for incomplete observations, and suffers from inherent uncertainties and ambiguities, which mainly stem from the latent causal relations, incomplete observations, and the reasoning itself. To address this, we propose a probabilistic model named Uncertainty-Guided Probabilistic Distillation Transformer (UPD-Trans) to model uncertainties for Visual Abductive Reasoning. In order to better discover the correct cause-effect chain, we model all the potential causal relations into a unified reasoning framework, thus both the direct relations and latent relations are considered. In order to reduce the effect of the stochasticity and uncertainty for reasoning: 1) we extend the deterministic Transformer to a probabilistic Transformer by considering those uncertain factors as Gaussian random variables and explicitly modeling their distribution; 2) we introduce a distillation mechanism between the posterior branch with complete observations and the prior branch with incomplete observations to transfer posterior knowledge. Evaluation results on the benchmark datasets, consistently demonstrate the commendable performance of our UPD-Trans, with significant improvements after latent relation modeling and uncertainty modeling.

## CCS CONCEPTS

- **Computing methodologies → Video summarization**.

## KEYWORDS

Visual Abductive Reasoning, Probabilistic Transformer, Uncertainty Modelling, Distillation Mechanism

**ACM Reference Format:**
Anonymous Author(s). 2024. Probabilistic Distillation Transformer: Modelling Uncertainties for Visual Abductive Reasoning. In *Proceedings of ACM Multimedia (ACMMM)*. ACM, New York, NY, USA, 9 pages. https://doi.org/XXXXXXX.XXXXXXX

## 1 INTRODUCTION

In recent years, deep neural networks have achieved significant success in various computer vision tasks [14, 17]. Although the perception abilities of current artificial vision systems have been approaching and even surpassing those of humans, they still lack the advanced cognitive understanding required for effectively reasoning

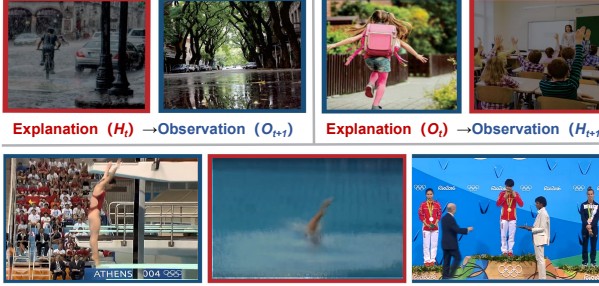

**Figure 1: Visual abduction reasoning aims to find the most plausible explanation for incomplete observations.**

about complex and dynamic real-world scenarios. This limitation poses a major obstacle to gaining a deeper understanding of the real world. Therefore, the CV community are paying growing attention to the problem of visual abductive reasoning (VAR) [15, 30], which aims to find the most plausible explanation for incomplete observations. Formally, the objective of AVR is to produce hypothesis ($H$) that is expected to best explain what happens before, after, or during the observation ($O$). As shown in Figure 1, if you observe $O_{t-1}$: "A woman walks on a springboard and jumps in the air." and $O_{t+1}$: "She stands on the podium waving her hands to the audience.", you can imagine $H_t$: "She then flips and dives in the water with a small splash." even without observing $O_t$.

To a certain extent, VAR resembles dense video captioning task [11, 27], but the main distinction arises from the necessity of reasoning and describing unobservable event conditioning on the past and/or future event. Furthermore, although VAR task shares similarities with video prediction task [8, 21], their main difference lies in the requirement for prediction at a higher semantic level, as opposed to mere low-level pixels, features, and states. Therefore, the VAR task is more challenging than vanilla vision-language (VL) tasks (*e.g.*, captioning) and vanilla vision (CV) tasks (*e.g.*, video prediction). Specifically, there are two limitations: 1) VAR suffers from the issue of incomplete observation and missing evidence chains. So it requires to capture more potential causal relations to provide go beyond visual evidences and unobservable latent evidences. 2) VAR suffers from inherent uncertainties, which mainly stem from the latent causal relations, incomplete observations, and the reasoning itself. For the 3-*th* example in Figure 1, only observing the past event cannot determine whether the athlete performed well or poorly. And after observing the future event "she standing on the podium", we can reasoning what happens. For the 1-*th* example in Figure 1, it is difficult to determine whether the ground is wet because of rain or because a water truck has sprayed water. It shows that this reasoning

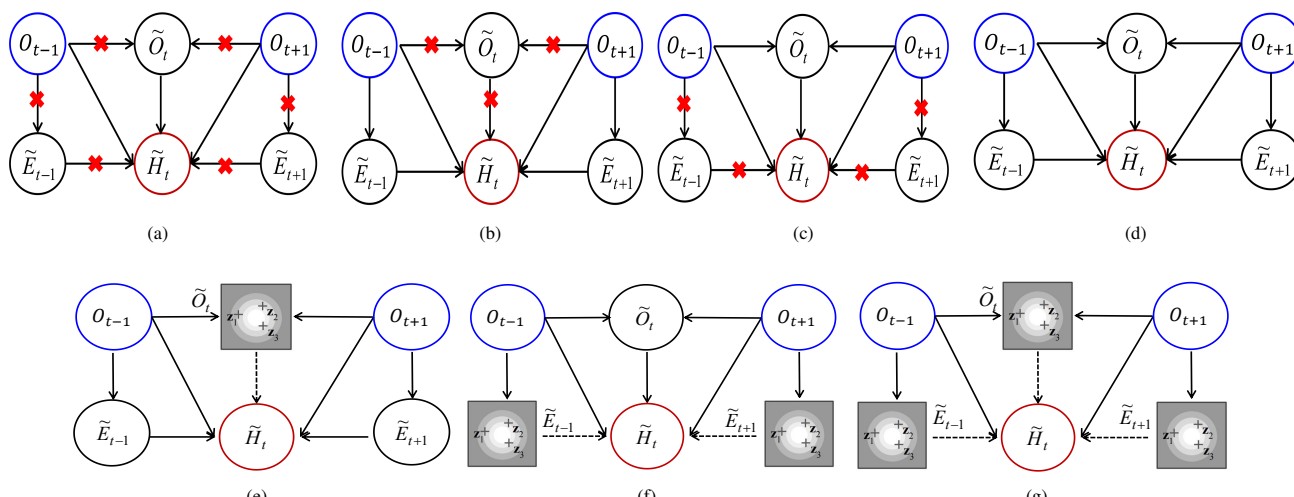

**Figure 2: The reasoning framework with different causal relation graphs: (a) Model with direct causal relations; (b)-(d) Model with direct and latent relations; (e)-(g) Uncertainty model with direct and latent relations. Observable $O_{t-1}/O_{t+1}$ and unobservable $\tilde{O}_t$ are treated as premise and explanation events, respectively; $\tilde{E}_{t-1}/\tilde{E}_{t+1}$ and $\tilde{H}_t$ are the descriptive sentences for the premise and explanation events, respectively.**

process is fraught with uncertainties, and it is imperative to consider all relevant relations to enhance the quality of reasoning.

To tackle the two limitations, we introduce a new probabilistic paradigm of VAR, named Uncertainty-Guided Probabilistic Distillation Trans-former (UPD-Trans) to model uncertainties and latent relations. 1) In order to capture all the potential causal relations and discover the correct cause-effect chain, we model all of the observable event observations (*i.e.*, Figure 2(a)), the unobservable event observations (*i.e.*, Figure 2(c)) and the unobservable event descriptions (*i.e.*, Figure 2(b)) into a unified reasoning framework (*i.e.*, Figure 2(d)), thus both the direct and the latent visual/linguistic relations can be considered. 2) In order to reduce the effect of the uncertainty for reasoning: firstly, we extend the deterministic Transformer to a probabilistic Transformer by considering those uncertain factors as Gaussian random variables and explicitly modeling their distribution as shown in Figure 2(e)-Figure 2(g); secondly, we incorporate a prior branch with incomplete observations and a posterior branch with complete observations, combined with a distillation module to transfer posterior knowledge for the prior branch.

## 2 RELATED WORKS

**Dense Video Captioning.** Our work may be, to some degree, similar to dense video captioning (DVC) [24], where the difference is DVC aims at captioning for observable event and VAR aims at captioning for unobservable event. DVC is to provide a detailed description with multiple sentences for all the events in an untrimmed video. In [25], a new MFT is proposed to generate paragraph descriptions that can preserve the story flow while being coherent and concise by assembling temporally localized descriptions. To alleviate the limitations of RNNs, an end-to-end DVC model with a masked transformer is proposed in [31], which utilizes a differentiable masking scheme to

guarantee the coherence among blocks. Memory-Augmented Recurrent Transformer (MART) [11] designed a memory module to augment the transformer architecture, which generates a highly summarized memory state to help better prediction of the next sentence. In [23], the author proposes a simple and effective end-to-end DVC framework (similar to DETR), which enhances the coherence and readability of predicted captions by accurately segmenting the video into multiple event segments based on the overall understanding of video content, by overlapping event counters on the top of the Transformer based decoder. A multi-modal single-stage dense event captioning model called Vid2Seq [27] is proposed, whose architecture enhances a language model by incorporating unique time tokens, enabling it to effortlessly predict event boundaries and textual descriptions within the same output sequence. A novel task of dense video captioning focusing on the generation of textual commentaries anchored with single timestamps is proposed in [19].

**Visual Prediction.** Our work is also moderately pertinent to video/feature prediction [18], where the difference is visual prediction aims at forecasting future frames, features or motion trajectories, and VAR aims at capturing the between-state changes to infer the explanations of missing parts. In [8], a Dynamic Multi-scale Voxel Flow Network (DMVFN) is proposed with a differentiable routing module that can effectively perceive the motion scales of video frames. A Prediction Conditional Generative Adversarial Network (Prediction-CGAN) is proposed in [26] for predicting action, which shares information between completely observed and partially observed videos. A novel next-frame video prediction method called PreCNet [21] is proposed for those images from an urban environment recorded from a car-mounted camera. In [10], a powerful framework for dense visual predictions based on the conditional diffusion pipeline is proposed with a "noise-to-map" generative paradigm. ViP3D [5] is proposed as the first fully differentiable

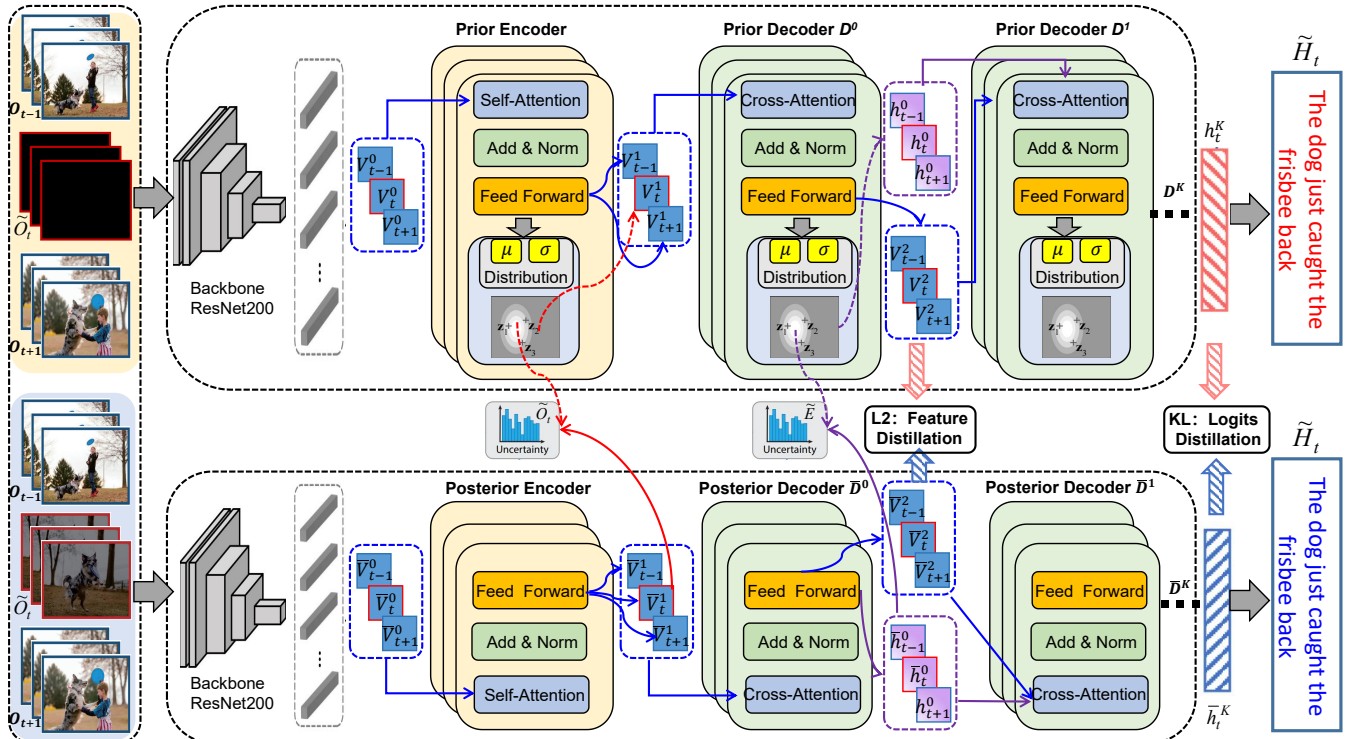

**Figure 3: Illustration of the proposed Uncertainty-Guided Probabilistic Distillation Transformer (UPD-Trans), consisting of a prior branch and a posterior branch with a distillation module and a uncertainty modeling module.**

vision-based trajectory prediction approach, which uses sparse agent queries to detect, track, and predict.

**Visual Abductive Reasoning.** Visual Abductive Reasoning [13] is an emerging vision-language (VL) topic aims at teaching the machine with the ability of inferring or reasoning from incomplete observations, and currently it mainly focus on image area. A dataset of SHERLOCK [7] is proposed for testing machine capacity for abductive reasoning beyond literal image contents. A new Regional Prompt Tuning with a Dual-Contrastive Loss is proposed in [28] for image abductive reasoning, where the relevant facts about inferences are not directly visible in the input images. In [12], the task of Causal-VidQA is proposed to facilitate deeper video understanding, which includes four types of questions ranging from scene description (description) to evidence reasoning (explanation) and commonsense reasoning (prediction and counterfactual). In [30], the model aims at figuring out what is the most plausible sequence of steps to achieve the goal, especially for those step-by-step instructional videos. Actually, the problem of visual abductive reasoning for incomplete video is first defined in [15] and introduce the first dataset for abductive reasoning in visual daily scenarios. In fact, VAR falls under the broader category of visual reasoning tasks [3, 4], and VAR task is more challenging. On the one hand, VAR seeks to reason from incomplete or unobservable inputs, whereas VR involves inferring from complete inputs that are not directly visible and directly achieved. On the other hand, unlike the open-ended VAR, although VR is also to infer the most plausible explanation for

visual observations, it is a hard hypothesis mining algorithm that is treated as a classification problem, akin to a multi-choice VQA task.

## 3 METHODOLOGY

### 3.1 Problem Definition

Given a video $\mathcal{V} = \{O_1,...,O_{t-1},\tilde{O}_t,O_{t+1},...O_T\}$ with $T$ temporally ordered events, the task of visual abductive reasoning is required to reason about the most likely explanation $\{\tilde{H}_t\}$ for the explanation events, where the observable premise events $\{O_1,...,O_{t-1},O_{t+1},...O_T\}$ and the unobservable explanation event $\{\tilde{O}_t\}$ are logically related.

$$P(\tilde{H}_t|\mathcal{V}) = \prod_l P(w_l|w_{<l},\mathcal{V}) = \prod_l P(w_l|w_{<l},O_{<t},O_{>t}) \quad (1)$$

where $\tilde{H}_t = \{w_1, w_2, ..., w_L\}$ denotes sentence that describe the content of the $t$-th explanation events, and $w_l$ is the $l$-th word. It is worth mentioning that the explanation event $\{\tilde{O}_t\}$ is unobservable, and the original representation of $\{\tilde{O}_t\}$ is obtained by setting its all values zero, so $\mathcal{V} = \{O_{<t},\tilde{O}_t,O_{>t}\} = \{O_{<t},O_{>t}\}$. To model $P(\tilde{H}_t|\mathcal{V})$ comprehensively, both the direct and the latent causal relations should be considered. The reasoning framework with different causal relation graphs as shown in Figure.2.

1) **The model with direct causal relations** is to directly reason $\tilde{H}_t$ from its past and future observable event observations, $P(\tilde{H}_t|\mathcal{V}) = P(\tilde{H}_t|O_{<t},O_{>t})$, as shown in Figure.2(a). The limitation

is that those potential relations that are most relevant to the inferred results have been overlooked.

2) **The model with direct and latent causal relations** is to reason $\tilde{H}_t$ conditioning on all of the possible factors, including the observable event observations $\{O_{<t}, O_{>t}\}$, the unobserved event observations $\tilde{O}_t$, the past and future event explanations $\{\tilde{E}_{<t}, \tilde{E}_{>t}\}$, thus $P(\tilde{H}_t|\mathcal{V}) = P(\tilde{H}_t|\tilde{E}_{<t}, \tilde{E}_{>t}, \tilde{O}_t, O_{<t}, O_{>t})$, as shown in Figure.2(d). The limitation is that although beneficial, those potential relations are unobservable and need to be inferred itself, introducing a certain degree of uncertainty and ambiguity.

3) **The uncertainty model with direct and latent causal relations.** Our UPD-Trans is formulated as Eq.2, for both latent relation modelling and uncertainty modelling.

$$
\begin{aligned}
P(\tilde{H}_t|\mathcal{V}) &= P(\tilde{H}_t|O_{<t}, O_{>t}) \\
&= \sum_{\tilde{E}_{<t}} \sum_{\tilde{E}_{>t}} P(\tilde{H}_t, \tilde{E}_{<t}, \tilde{E}_{>t}|O_{<t}, O_{>t}) \\
&= \sum_{\tilde{O}_t} \sum_{\tilde{E}_{<t}} \sum_{\tilde{E}_{>t}} P(\tilde{H}_t, \tilde{E}_{<t}, \tilde{E}_{>t}, \tilde{O}_t|O_{<t}, O_{>t}) \\
&\approx \sum_{\tilde{O}_t} \sum_{\tilde{E}_{<t}} \sum_{\tilde{E}_{>t}} P(\tilde{H}_t|\tilde{E}_{<t}, \tilde{E}_{>t}, \tilde{O}_t, O_{<t}, O_{>t}) \\
&\quad \times P(\tilde{O}_t|O_{<t}, O_{>t}) \times P(\tilde{E}_{<t}, \tilde{E}_{>t}|O_{<t}, O_{>t})
\end{aligned}
\tag{2}
$$

By explicitly modeling the distribution of those potential factors and considering their all possible values, we extend the deterministic Transformer to a probabilistic Transformer in order to reduce the effect of uncertainty. As shown in Figure.2(g), it can be treated as a general framework for cases of Figure.2(a)-Figure.2(f), where each of them is an approximation or particular case of our UPD-Trans.

## 3.2 Model Architecture

The overall pipeline of the proposed Uncertainty-Guided Probabilistic Distillation Transformer (UPD-Trans) is shown in Figure.3. We propose the multi-modal mixture of encoder-decoder, which consists of four sub-modules: 1) A prior reasoner consists of a observation-predicting encoder and a cascaded-reasoning decoder, whose input is the incomplete observations, and only this prior branch is used for inference; 2) A posterior reasoner is implemented with the similar architecture to the prior branch, whose input is the complete observations, and this prior branch is only used for training; 3) A distillation module between the prior branch and posterior branch is introduced to distill and transfer posterior knowledge for the prior branch. 4) A uncertainty modeling module is proposed to model the uncertainties involved in the unobservable event observations $\tilde{O}_t$ and the inferred past and future event descriptions $\{\tilde{E}_{<t}, \tilde{E}_{>t}\}$.

**Prior Reasoner and Posterior Reasoner.**

1) Observation-Predicting Encoder: its purpose is to make use of contextual information from past and/or future observable events to enhance their own representations and also predict a meaningful representation for the most probable explanatory hypothesis. The original representation for the prior branch is $\mathbf{V}^0_{prior} = \{v_1^0, ..., v_{t-1}^0, \tilde{v}_t^0, v_{t+1}^0, ...v_T^0\}$, $\mathbf{V}^0_{prior} \in R^{T \times d}$, where the representation of unobservable explanation event is masked, namely, $\tilde{v}_t^0 = \mathbf{0}^{1 \times d}$; while the original representations for the posterior branch are all observable $\mathbf{V}^0_{post} = \{\bar{v}_1^0, ..., \bar{v}_{t-1}^0, \bar{v}_t^0, \bar{v}_{t+1}^0, ...\bar{v}_T^0\}$.

$$
\mathbf{V}^1 = soft\max(\mathbf{V}^0\mathbf{W}^q(\mathbf{V}^0\mathbf{W}^k)^T)\mathbf{V}^0\mathbf{W}^v
\tag{3}
$$

where $\mathbf{V}^0 \in \{\mathbf{V}^0_{prior}, \mathbf{V}^0_{post}\}$. After obtaining the $\mathbf{V}^1 \in \{\mathbf{V}^1_{prior}, \mathbf{V}^1_{post}\}$ via the above attention operation, we adopt a hypothesis prediction based optimization criterion, to give clearer guidance to the prior encoder and improve reasoning.

$$
L_{pre} = ||f_{proj}(\tilde{v}_t^1) - f_{proj}(\bar{v}_t^1)||_2
\tag{4}
$$

where $\tilde{v}_t^1$ is the predicted representation for the unobservable explanation event obtained from the prior branch with incomplete inputs, $\bar{v}_t^1$ is the enhanced representation for the unobservable explanation event obtained from the posterior branch with complete inputs, and $f_{proj}$ is a project layer with several MLPs. Thus, $P(\tilde{O}_t|O_{<t}, O_{>t})$ is modeled via the observation-predicting encoder. Since $\tilde{v}_t^1$ is the representation of $\tilde{O}_t$, we do not differentiate between the two in this paper. Specifically, the encoder is implemented as two Transformer encoder blocks with 12 attention heads and a dimension of 768 hidden representation.

2) Cascaded-Reasoning Decoder: its purpose is to model $P(\tilde{H}_t|\tilde{E}_{<t}, \tilde{E}_{>t}, \tilde{O}_t, O_{<t}, O_{>t})$, and generate a series of descriptive sentences for premise events $\{\tilde{E}_1^k, ..., \tilde{E}_{t-1}^k, \tilde{E}_{t+1}^k, ..., \tilde{E}_T^k\}_{k=0}^K$ and explanation event $\{\tilde{H}_t^k\}_{k=0}^K$. The total number of cascaded decoders is $K$ and $\tilde{H}_t^K = \tilde{H}_t = \{w_1, w_2, ..., w_L\}$ is the final output. The advantages of the cascaded reasoning have two folds: on the one hand, the $\{\tilde{E}_1^k, ..., \tilde{E}_{t-1}^k, \tilde{H}_t, \tilde{E}_{t+1}^k, ..., \tilde{E}_T^k\}_{k=0}^K$ is refined step-by-step, gradually enhancing the reasoning ability; on the other hand, it models the $P(\tilde{E}_{<t}, \tilde{E}_{>t}|O_{<t}, O_{>t})$ and provides a way to capture the latent relations between the $\tilde{H}_t^k$ and $\tilde{E}^{k-1} = \{\tilde{E}_{>t}^{k-1}, \tilde{E}_{<t}^{k-1}\}$.

Specifically, each decoder module is implemented as two multi-modal masked Transformer decoder blocks with 12 attention heads and a dimension of 768 hidden representation. By integrating visual and linguistic representations as input, $D^0$ performs cross-modal reasoning, thereby facilitating enhanced event representation $\mathbf{V}^1 \to \mathbf{V}^2$ and updating the visual-linguistic state for each event, *i.e.*, $\mathbf{H}_{1:T}^0 = \{\mathbf{H}_1^0, \mathbf{H}_2^0, ..., \mathbf{H}_T^0\}$ and each $\mathbf{H}_t^0 \in R^{L \times d}$.

$$
[\mathbf{V}^2, \mathbf{H}_{1:T}^0] = D^0([\mathbf{V}^1, \mathbf{H}_{1:T}])
\tag{5}
$$

where each $\mathbf{H}_t \in R^{L \times d}$ is a set of $L$ words embedding for the ground-truth description of each event and it is recurrently generated during inference. Then the initial description of explanation event $\tilde{H}_t^0$ or premise event $\tilde{E}_n^0 (n \neq t)$ is generated via $D^0$, where a captioning head is adopted to map the visual-linguistic state $\mathbf{H}_t^0$ into word distribution.

$$
P(\tilde{H}_t^0|\mathcal{V}) = P(\tilde{H}_t^0|\mathbf{V}^1) = \prod_l P(w_l^0|w_{<l}^0, \mathbf{H}_t^0)
\tag{6}
$$

Similar for those premise events, that is $P(\tilde{E}_n^0|\mathcal{V}) = \prod_l P(w_l^0|w_{<l}^0, \mathbf{H}_n^0), n \neq t$.

The novel multi-step description refinement process involves the inclusion of multiple Transformer decoder blocks over the initial description $D^0$. To be more precise, our entire refinement procedure can be recursively defined as,

$$
[\mathbf{V}^{k+2}, \mathbf{H}_{1:T}^k] = D^k([\mathbf{V}^{k+1}, \mathbf{H}_{1:T}, h_{1:T}^{k-1}])
\tag{7}
$$

$$
P(\tilde{H}_t^k|\mathcal{V}) = P(\tilde{H}_t^k|\mathbf{V}^{k+1}, h_{1:T}^{k-1}) = \prod_l P(w_l^k|w_{<l}^k, \mathbf{H}_t^k)
\tag{8}
$$

where $D^k$ denotes the $k$-th refinement module; $h_t^{k-1} \in R^d$ refers to a condensed representation of $\mathbf{H}_t^{k-1} \in R^{L \times d}$: $h_t^{k-1} = \max pool(\mathbf{H}_t^{k-1})$, which implicitly contains the evidences of $(\tilde{E}_{<t}, \tilde{H}_t, \tilde{E}_{>t})$. By adopting this cascaded approach, each $D^k$ can make full use of previously generated descriptions of explanation event $\tilde{H}_t^{k-1}$ for refinement and inter-sentential relations from past and future premise events $(\tilde{E}_{<t}, \tilde{E}_{>t})$ for comprehensive relation modeling.

**Distillation between Prior and Posterior.** Since the input of the posterior branch is complete observations, its reasoning is more confident with less uncertainty than the prior branch with incomplete observations. Therefore, a distillation module between the prior branch and posterior branch is introduced to distill and transfer posterior knowledge for the prior branch. By the prior reasoner and posterior reasoner, we obtain the visual representations $\{\mathbf{V}_{prior}^k = v_{1:T}^k, \mathbf{V}_{post}^k = \bar{v}_{1:T}^k\}$ and word distribution $\{P_{prior}^k(\tilde{H}_t^k|\mathcal{V}), P_{post}^k(\tilde{H}_t^k|\mathcal{V})\}$ for each refinement step. And feature distillation and logits distillation are both adopted.

1) Feature Distillation:

$$L_{distill1} = \sum_{k=1}^{K} \sum_{t=1}^{T} ||f_{proj}(v_t^k) - f_{proj}(\bar{v}_t^k)||_2 \qquad (9)$$

2) Logits Distillation:

$$L_{distill2} = \sum_{k=1}^{K} KL(P_{prior}^k(\tilde{H}_t^k|\mathcal{V})||P_{post}^k(\tilde{H}_t^k|\mathcal{V})) \qquad (10)$$

**Probabilistic Modeling for Uncertainty.** In the conventional Transformer, predicted observations $\tilde{O}$ and inferred descriptions $\tilde{E}$ are computed deterministically by Eq.3 and Eq.8 respectively, which is referred as deterministic output. However, these inferred predictions that are used as inputs for the subsequent decoder are likely to amplify the uncertainty and ambiguity contained in them, resulting in cumulative errors. Therefore, we extend the Transformer model to a probabilistic Transformer by considering those uncertain factors as Gaussian random variables and explicitly modeling their distribution via adding a probabilistic feed-forward layers. There are mainly two uncertain factors, namely $\tilde{O}$ and $\tilde{E}$, where we actually conduct the uncertainty modeling for their representations $\tilde{v}_t^1$ and embedding state $h_t^k$, respectively.

1) Uncertainty Modeling for Predicted Observations. Instead of conducting a deterministic prediction with large uncertainty for the representation of $\tilde{O}_t$, we model its uncertainty and assume $\tilde{v}_t^1$ follows a Gaussian distribution, thus we can achieve $P(\tilde{O}_t|O_{<t}, O_{>t})$. The mean and variance of this Gaussian distribution is computed using $\tilde{v}_t^1$ through a multi-layer perception, *i.e.*, $f_\mu, f_\sigma = Linear(Act(Linear(.)))$. Although this distribution is unknown, we can constraint its sampling values to approach to $\bar{v}_t^1$ by extending Eq.4 to Eq.12. To train these probabilistic layers through the gradient descent, we adopt the reparameterization trick to perform the forward process of the probabilistic output.

$$\tilde{v}_t^1 : z_o \sim N(\mu_o, \sigma_o^2 I), \mu_o = f_\mu(\tilde{v}_t^1), \sigma_o = f_\sigma(\tilde{v}_t^1) \qquad (11)$$

$$L_{uo} = ||f_{proj}(z_o) - f_{proj}(\tilde{v}_t^1)||_2 \qquad (12)$$

2) Uncertainty Modeling for Inferred Descriptions. Similarly, instead of conducting a deterministic prediction with large uncertainty

---

**Algorithm 1** Training Procedure for the UPD-Trans

1: **Input**: $D = \{\mathcal{V}^n, G_t^n\}_{n=1}^N$
2: **Output**: $\Theta$
3: Initialize parameters of $\Theta$.
4: **for** $n \in [1, N]$ in $\{\mathcal{V}^n, G_t^n\} \in D$ **do**
5:    Obtain the original features $\mathbf{V}_{prior}^0, \mathbf{V}_{post}^0 \in R^{T \times d}$
6:    Obtain $\{\mathbf{V}_{prior}^1, \mathbf{V}_{post}^1\}$ via Encoder using Eq.3
7:    Sample $\tilde{v}_t^1$ using Eq.11
8:    **for** $k \in [0, K]$ of Cascaded-Reasoning Decoder **do**
9:       Update $\{\mathbf{V}_{prior}^k = v_{1:T}^k, \mathbf{V}_{post}^k = \bar{v}_{1:T}^k\}$ and $\{\mathbf{H}_{prior}^k = h_{1:T}^k, \mathbf{H}_{post}^k = \bar{h}_{1:T}^k\}$ using Eq.7 or Eq.5
10:       Compute $\{P_{prior}^k(\tilde{H}_t^k|\mathcal{V}), P_{post}^k(\tilde{H}_t^k|\mathcal{V})\}$ using Eq.8
11:       Sample $h_t^k$ using Eq.13
12:    **end for** $k$
13:    Optimize $\Theta$ by optimizing Eq.14+Eq.12+Eq.15+Eq.16 +Eq.9+Eq.10
14: **end for** $n$
15: **return** $\Theta^*$

---

for the embedding state of $\{\tilde{E}_{>t}, \tilde{E}_{<t}\}$, we model its uncertainty and assume $h_t^k$ follows a Gaussian distribution, thus we can achieve $P(\tilde{E}_{<t}, \tilde{E}_{>t}|O_{<t}, O_{>t})$. The mean and variance of this Gaussian distribution is computed using $h_t^k$ through a MLP. Although this distribution is unknown, we can constraint its sampling values to approach to $\bar{h}_t^k$ generated via the completed posterior branch by Eq.14.

$$h_t^k : z_e \sim N(\mu_e, \sigma_e^2 I), \mu_e = f_\mu(h_t^k), \sigma_e = f_\sigma(h_t^k) \qquad (13)$$

$$L_{ue} = ||f_{proj}(z_e) - f_{proj}(\bar{h}_t^k)||_2 \qquad (14)$$

By the uncertainty modeling, we extend $P(\tilde{H}_t|\mathcal{V}) = P(\tilde{H}_t|\tilde{E}_{<t}, \tilde{E}_{>t}, \tilde{O}_t, O_{<t}, O_{>t})$ to Eq.2. Since directly computing the integration of $\tilde{O}_t$ and $\{\tilde{E}_{<t}, \tilde{E}_{>t}\}$ is intractable, we sample $\tilde{v}_t^1$ and $h_t^k$ from Eq.11 and Eq.13 for $M$ times to approximate this probabilistic distribution. The validation results from subsequent experiments will provide confirmation regarding of the sampling number.

### 3.3 Optimization and Inference

For each training samples in $D = \{\mathcal{V}^n, G_t^n\}_{n=1}^N$, its ground-truth is $G_t^n = \{\hat{w}_1^n, \hat{w}_2^n, ..., \hat{w}_L^n\}$ and its observations $\mathcal{V}^n = \{O_{<t}^n, \tilde{O}_t^n, O_{>t}^n\}$, where $\tilde{O}_t^n = \mathbf{0}$ for prior branch and $\tilde{O}_t^n = O_t^n$ for posterior branch. For the main task of reasoning, we propose to use the negative log-likelihood (NLL) loss to train the two branches.

$$L_{NLL}^{prior} = -\sum_{k=1}^{K} \sum_{l=1}^{L} P(\hat{w}_l|w_{<l}^k, \mathbf{H}_t^k) \qquad (15)$$

$$L_{NLL}^{post} = -\sum_{k=1}^{K} \sum_{l=1}^{L} P(\hat{w}_l|w_{<l}^k, \bar{\mathbf{H}}_t^k) \qquad (16)$$

Additionally, the loss of Eq.9 and Eq.10 is used for knowledge distillation, and the loss of Eq.14 and Eq.12 is used for uncertainty learning. The whole training process is summarized as Algorithm 1.

**Inference Process:** The posterior branch with complete observable events is incorporated solely during the training phase to regulate the uncertainty distribution of the prior branch and to furnish

**Table 1: Comparisons with the state-of-the-art methods on VAR dataset.**

| Methods | Encoder-Decoder | BLEU@4 | METEOR | ROUGE | CIDEr | BERT-S |
|---------|-----------------|--------|--------|-------|-------|--------|
| Human | - | 11.35 | 19.36 | 36.92 | 147.79 | 40.59 |
| VTrans [31] | Trans.-Trans. | 0.71 | 6.92 | 19.12 | 7.11 | 22.13 |
| MFT [25] | RNN-RNN | 1.81 | 7.16 | 19.16 | 17.67 | 25.90 |
| Trans-XL [2] | Trans.-Trans. | 2.96 | 7.51 | 20.94 | 24.54 | 27.23 |
| MART [11] | Trans.-Trans. | 2.86 | 7.47 | 20.87 | 24.05 | 27.77 |
| PDVC [23] | Trans.-Trans. | 3.00 | 8.54 | 20.71 | 25.14 | 27.80 |
| REASONER [15] | Trans.-RNN | 3.44 | 9.05 | 22.89 | 30.75 | 30.64 |
| **UPD-Trans** | Trans.-Trans. | **5.40** | **11.16** | **25.62** | **41.66** | **30.80** |

the prior branch with more reliable information. Specifically, during training all events within the video are observable by assigning $\tilde{O}_t = O_t$ within the set $\mathcal{V} = \{O_1, ..., O_{t-1}, \tilde{O}_t, O_{t+1}, ...O_T\}$. Conversely, during inference, only the prior branch with incomplete observable events is used, where the explanation event is treated as unobservable by initializing the original representation of $\tilde{O}_t$ with zeros.

## 4 EXPERIMENTS

### 4.1 Dataset and Implementation Details

**Dataset.** In this paper, we evaluate our proposed UPD-trans on VAR [15], and it is the first dataset for abductive reasoning in visual daily scenarios. The VAR dataset comprises a total of 8606 data examples, which were gathered from 3718 distinct videos. Each video includes 4.17 events with an average duration of 37.8 seconds on average. Consequently, the dataset includes a total of 15K descriptive sentences, with an average length of 13.5 words. The dataset is split into training, validation, and testing partitions, containing 7053, 460, and 1093 videos, respectively.

**Implementation Details.** Our model is implemented using PyTorch and trained on a server equipped with two P40 GPUs. Two Transformer encoder blocks are implemented as the encoder of UPD-Trans and two Transformer masked decoder blocks are implemented as each decoder module. The hidden size of the representation is set to $n = 768$, with 12 attention heads employed. For cascaded reasoning, a total of $K = 2$ decoders are stacked via cross-validation. We sample 50 frames for each event in a uniform manner and combine their features to form the original event representation, which is extracted by ResNet200 [6]/BN-Inception [9] with a dimension of 3072. Descriptive sentences for each event are padded or truncated into 20 words. During training, we use the Adam optimizer with an initial learning rate of 1e-4 and a small decaying rate. Furthermore, dropout regularization is applied to prevent overfitting.

### 4.2 Compare to the State-of-the-Art Methods

We conduct comparisons to the state-of-the-art methods on the test set of VAR dataset and show the automatic evaluation results of visual abductive reasoning in Table 1. The comparative methods include some methods (*i.e.*, REASONER [15]) specifically designed for visual abductive reasoning task and others for dense video captioning (*i.e.*, VTrans [31], MFT [25], Trans-XL [2], MART [11], PDVC [23]), as visual abductive reasoning is still a relatively new

task and there aren't many dedicated methods proposed for it currently. Additionally, the human performance is provided as the upper bound of VAR performance, which is achieved by ten volunteers. For the evaluation metrics, five widely used metrics in image/video captioning are adopted, including BERTScore [29], BLEU4 [20], METEOR [1], ROUGE-L [16] and CIDEr [22]. We can draw the following observations: 1) The comparisons show that our UPD-Trans achieves much better performance for visual abductive reasoning than the SOTA models. Compared to the second-best method [15], we achieve a gain of 1.96, 2.11, 2.73, 10.91 and 0.16, and dramatically higher than other DVC methods. 2) It is evident that there is a significant gap between captioning for observable event and reasoning for unobservable event, where existing DVC models has extremely poor performance on reasoning beyond observation. 3) Our UPD-Trans outperforms many famous video-language models, while still being far behind human performance, which indicates there needs further research for this new reasoning task.

### 4.3 Ablation Studies

**Effectiveness of Each Component:** To evaluate the effectiveness of each individual component, we evaluate several variants of our method on VAR dataset as demonstrated in Figure 2(a)-Figure 2(g). The results are shown in Table 2, and all components in UPD-Trans contribute to the performance of visual abduction reasoning in general. The detail analysis for each component is given as following: 1) By simultaneously modeling direct and latent causal relations, we achieve a gain of 0.81, 0.79, 4.7, 1.94 and 3.36, which demonstrates the effective of incorporating Latent($\tilde{O}$) and Latent($\tilde{E}$) for reasoning. 2) By conducting probabilistic modeling for the latent relations, it promotes all the five scores and dramatically promotes the CIDEr score from 35.58 to 40.19, which demonstrates the effective of the probabilistic Transformer for uncertainty modeling. 3) By incorporating a distillation module between the two branches, it boosts the CIDEr score by 3.1, which demonstrates the posterior branch with complete observable data indeed benefits for reducing uncertainty in the prior branch without complete observable data.

**Effectiveness of the Prior Branch and Posterior Branch:** We conducted several evaluations to demonstrate the effectiveness of both the prior branch and the posterior branch, as presented in Table 3. In UPD-Trans, the functions of the posterior branch are as follows: 1) For uncertainty modeling, the posterior branch is utilized to regulate the uncertainty distribution of predicted observations ($\tilde{O}$) and inferred descriptions ($\tilde{E}$) in the prior branch. 2) For the distillation

**Table 2: The effectiveness of each individual component of UPD-Trans evaluated on VAR dataset.**

| Latent($\tilde{O}$) | Latent($\tilde{E}$) | Uncertainty($\tilde{O}$) | Uncertainty($\tilde{E}$) | Feature Distill | Logits Distill | VAR | | | | |
|---|---|---|---|---|---|---|---|---|---|---|
| | | | | | | M | B@4 | C | R | BERT |
| ✗ | ✗ | ✗ | ✗ | ✗ | ✗ | 9.65 | 4.13 | 30.88 | 21.96 | 26.94 |
| ✓ | ✗ | ✗ | ✗ | ✗ | ✗ | 10.18 | 4.57 | 33.09 | 22.86 | 29.04 |
| ✗ | ✓ | ✗ | ✗ | ✗ | ✗ | 10.53 | 4.75 | 32.84 | 23.49 | 28.84 |
| ✓ | ✓ | ✗ | ✗ | ✗ | ✗ | 10.46 | 4.92 | 35.58 | 23.90 | 30.30 |
| ✓ | ✓ | ✓ | ✗ | ✗ | ✗ | 10.72 | 5.23 | 39.42 | 24.65 | 30.54 |
| ✓ | ✓ | ✗ | ✓ | ✗ | ✗ | 10.78 | 4.80 | 39.18 | 24.78 | **30.87** |
| ✓ | ✓ | ✗ | ✗ | ✓ | ✗ | 10.88 | 5.25 | 38.17 | 25.08 | **30.87** |
| ✓ | ✓ | ✗ | ✗ | ✗ | ✓ | 10.95 | 5.05 | 37.88 | 25.02 | 30.30 |
| ✓ | ✓ | ✗ | ✗ | ✓ | ✓ | 10.74 | 5.22 | 38.68 | 25.46 | 30.52 |
| ✓ | ✓ | ✓ | ✓ | ✗ | ✗ | 10.73 | 5.12 | 40.19 | 24.54 | 30.76 |
| ✓ | ✓ | ✓ | ✓ | ✓ | ✓ | **11.16** | **5.40** | **41.66** | **25.62** | 30.80 |

**Table 3: The effectiveness of the prior and posterior branch.**

| Methods | M | B4 | C | R | Bert |
|---|---|---|---|---|---|
| Prior | 9.65 | 4.13 | 30.88 | 21.96 | 26.94 |
| Prior+$\tilde{E}$ | 10.53 | 4.75 | 32.84 | 23.49 | 28.84 |
| Prior+ U($\tilde{E}$) | 10.25 | 4.95 | 34.71 | 23.77 | 30.30 |
| Prior+ U($\tilde{E},\tilde{O}$) | 10.69 | 4.92 | 36.89 | 24.72 | 30.67 |
| Prior+Posterior+ U($\tilde{E},\tilde{O}$) | 10.73 | 5.12 | 40.19 | 24.54 | 30.76 |
| Prior+Posterior+Distill | 10.74 | 5.22 | 38.68 | 25.46 | 30.52 |
| UPD-Trans | **11.16** | **5.40** | **41.66** | **25.62** | **30.80** |
| Posterior (all observed) | 11.36 | 5.25 | 41.64 | 25.50 | 30.94 |
| UPD-Trans (all observed) | **11.56** | **5.70** | **44.39** | **25.74** | **30.97** |

**Table 4: Performance of visual abductive reasoning with different number of cascaded decoder on VAR dataset.**

| $D^K$ | METEOR | BLEU@4 | CIDEr | ROUGE | BERT-S |
|---|---|---|---|---|---|
| K=0 | 10.49 | 4.45 | 35.34 | 23.70 | 27.87 |
| K=1 | 10.72 | **5.40** | 39.81 | 24.05 | 30.21 |
| K=2 | **11.16** | **5.40** | **41.66** | **25.62** | **30.80** |
| K=3 | 10.51 | 5.10 | 38.47 | 24.21 | 30.09 |
| K=4 | 10.55 | 4.85 | 36.39 | 24.19 | 30.32 |
| K=5 | 10.60 | 5.19 | 37.07 | 24.56 | 30.38 |

**Table 5: Performance of visual abductive reasoning with different sampling numbers on VAR dataset.**

| M | METEOR | BLEU@4 | CIDEr | ROUGE | BERT-S |
|---|---|---|---|---|---|
| $\mu$ | 11.14 | 5.33 | 41.58 | 25.34 | 30.64 |
| 1 | 11.16 | 5.40 | 41.66 | **25.62** | 30.80 |
| 3 | 11.14 | 5.34 | 41.38 | 25.47 | 30.81 |
| 5 | **11.21** | **5.43** | **42.24** | 25.53 | 30.76 |
| 8 | 11.18 | 5.35 | 41.76 | 25.52 | **30.84** |
| 10 | 11.10 | 5.34 | 41.91 | 25.41 | 30.81 |

process, the posterior branch offers more reliable information to the prior branch, since all events in the posterior branch are observed. The evaluations in Table 3 confirm the effectiveness of these two functions, wherein the 'Prior+Posterior+ U($\tilde{E},\tilde{O}$)' outperforms 'Prior+ U($\tilde{E},\tilde{O}$),' and 'Prior+Posterior+Distill' achieves better results than 'Prior+$\tilde{E}$.' The final two lines can be considered an upper bound of this model, where all events are observed during testing and training, ensuring the accurate transfer of information from the posterior branch to the prior branch. Furthermore, even when all events are observed, uncertainties still exist, as seen in the comparison between 'Posterior (all observed)' and 'UPD-Trans (all observed).'

**Effectiveness of Cascaded Reasoning:** Table 4 presents the performance of visual abductive reasoning with different number of cascaded decoder on VAR dataset, denoted as $K = 0, 1, 2, 3, 4, 5$. When $K = 0$, we only employ the initial decoder $D^0$, yielding a CIDEr score of 35.34. By incorporating additional refinement decoder, the score significantly improves to 41.66. The improvement continues until $K > 2$, where the increasing trend eventually reaches a saturation point. Consequently, we adopt $K = 2$ as the default setting to strike a balance between performance and computational efficiency.

**Effectiveness of Sampling Numbers:** Table 5 presents the performance of visual abductive reasoning with different sampling numbers during inference on VAR dataset. When using $\mu$, we only

employ the mean value as the approximation of $\bar{v}_t^1$ and $h_t^k$, yielding a CIDEr score of 41.58. Due to multiple sampling iterations during the training process, the model's sensitivity to the sampling number during testing is not significant, and there is no noticeable difference in performance. Therefore, we default to sampling only once during testing, taking computational efficiency into account.

## 4.4 Qualitative Results

We showcase several visual abductive reasoning examples in Figure 4, which provide further evidence of our model's superior performance. Where the premise event is in blue and the explanation event is in red. It is obviously that our UPD-Trans is able to discover and correctly describe the cause-effect chain, to a certain extent, and hence generate a plausible hypothesis, i.e., "The person then lights a

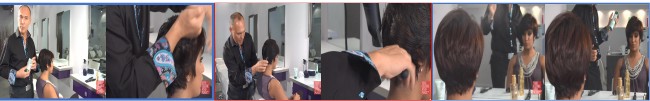

**Baseline:** [She then holds up a bottle of lotion and begins speaking to the camera while holding up the camera . ]
**UPD-Trans:** [He then shows off the hair and begins to blow dry the hair and then takes a brush . ]
**Groundtruth:** [Premise: A man is seen speaking to the camera while holding a bottle of lotion and then spread it throughout a woman's hair] [Explanation: He then brushes the hair and blow dries it while styling it into a certain look ] [Premise: He finishes the cut and dry and the woman sees herself in the mirror.]

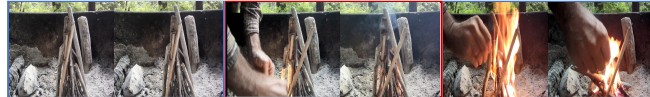

**Baseline:** [A man is standing in a pile of wood next to a pile of wood . ]
**UPD-Trans:** [The person then lights a fire and lights a piece of wood.]
**Groundtruth:** [Premise: A guy is trying to lite a pale of sticks in a round barrole.] [Explanation: He finally gets the pale of sticks lite and as the fire grows the pale of sticks fall. ] [Premise: The guy add sticks to keep the fire going.]

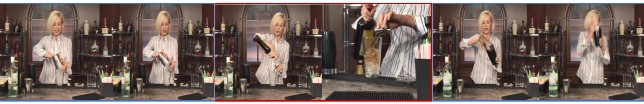

**Baseline:** [She pours some ice into a glass and shakes it up. ]
**UPD-Trans:** [She begins by mixing the ingredients into a glass and filling it.]
**Groundtruth:** [Premise: A woman pours ice into a glass.] [Explanation: She adds shots of alcohol to the glass. ] [Premise: She then pours it into another glass and shakes it.]

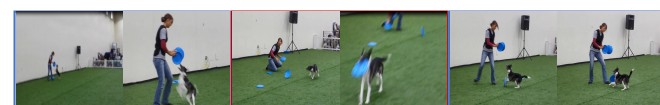

**Baseline:** [The man throws a frisbee to the dog. ]
**UPD-Trans:** [The man then begins to throw the frisbee and the dog catches it.]
**Groundtruth:** [Premise: Person is holgding blue frisbees and is plying with a dog in a closed field, the doing tricks while is trying to catch the frisbee.] [Explanation: The dog holds a frisbee on his mouth and starts running around the girl doing tricks. ] [Premise: The dog stands on her feet and waits for her to throw the frisbees.]

**Figure 4: Visualization of some VAR examples, including the examples generated by the baseline, our UPD-Trans and human annotation.**

fire and lights a piece of wood", that well explains the observable events, "The guy add sticks to keep the fire going". It is achieved by effectively incorporating potential latent relations and uncertainty modeling. In contrast, the performance of the baseline (REASONER [15] ) is unsatisfactory, which always repeat describe the observable premise events.

## 5 CONCLUSION

Different from the task of video captioning, visual abduction reasoning is more challenging as it requires to conduct reasoning beyond observation. In this paper, we propose a novel reasoning method called Uncertainty-Guided Probabilistic Distillation Transformer. In order to improve reasoning capability, UPD-Trans better discovers correct cause-effect chain by incorporating both direct and latent causal relations. To address the issue of uncertainty, we extend the deterministic Transformer to a probabilistic Transformer, which allows for modeling uncertainty in variables by the form of probabilistic distribution. Additionally, we utilize a posterior reasoner to distill and transfer more confirmed knowledge to the prior reasoner. As a result, our UPD-Trans can capture more latent relations and reduce uncertainty, leading to enhanced reasoning capability. We evaluate UPD-Trans on the new VAR dataset and demonstrate its remarkable performance.

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
