# OpenReview forum: "Probabilistic Distillation Transformer: Modelling Uncertainties for Visual Abductive Reasoning"
_acmmm.org/ACMMM/2024/Conference — MM2024 Poster_

### Official Review · Reviewer_MZB1 · 2024-05-24

**Rating:** 4
**Confidence:** 3

**Summary:**

The paper introduces the Uncertainty-Guided Probabilistic Distillation Transformer (UPD-Trans), a novel approach aimed at enhancing Visual Abductive Reasoning by effectively modeling uncertainties stemming from latent causal relationships and incomplete observations. The model integrates a dual-branch architecture with distillation between a prior and posterior model, and employs probabilistic modeling of uncertainties using Gaussian random variables. The evaluation on benchmark datasets demonstrates significant performance improvements over existing methods.

**Strengths:**

1. The UPD-Trans introduces a novel architecture that effectively handles the inherent uncertainties in visual abductive reasoning, employing a probabilistic transformer and a distillation mechanism, which is novel for the task.
2. The paper provides a thorough evaluation, including comprehensive ablation studies that clearly demonstrate the impact of each component of the model. The results show significant improvements over state-of-the-art models.

**Limitations:**

1. The paper focuses predominantly on specific datasets and scenarios. A broader discussion on the applicability of the model to diverse real-world scenarios could enhance its relevance.
2. The most recent method discussed in experiments was published in 2022.  To provide a more comprehensive analysis, it is essential to include more recent methods. I am particularly interested in assessing the performance of current large-scale models, such as Gemini and GPT-4, on this task.  While I understand the challenges of incorporating this during the rebuttal phase, could you possibly include some visualizations or subset performance?

**Suitability:**

3

---

### Official Review · Reviewer_ehC3 · 2024-05-24

**Rating:** 4
**Confidence:** 2

**Summary:**

The paper proposes a probabilistic model named Uncertainty-Guided Probabilistic Distillation Transformer (UPD-Trans) to model
uncertainties for Visual Abductive Reasoning. Specifically, the proposed method i) extends the deterministic Transformer to a probabilistic Transformer by considering those uncertain factors as Gaussian random variables and explicitly modeling their distribution and ii) introduces a distillation mechanism between the posterior branch with complete observations and the prior branch with incomplete observations to transfer posterior knowledge. Extensive empirical results on the benchmark datasets demonstrate the commendable performance of
the proposed method.

**Strengths:**

+ The paper proposes to transfer the complete observation knowledge with distillation between prior and posterior reasoners, which is interesting.
+ Extensive experiments demonstrate the effectiveness of this approach.
+ The ablative studies are thorough.
+ The presentation is clear and easy to follow.

**Limitations:**

- Some minor issues like typos in line 88: AVR -> VAR.
- Admittedly, I am not very familiar with this domain, and would like to hear about other reviewers' opinions regarding the limitations.

**Suitability:**

3

---

### Official Review · Reviewer_wDtt · 2024-05-26

**Rating:** 3
**Confidence:** 3

**Summary:**

This work introduces the Uncertainty-Guided Probabilistic Distillation Transformer (UPD-Trans) to tackle the challenges of visual abduction reasoning. UPD-Trans employs a dual approach to integrate overt and underlying causal connections, and extend the traditional deterministic Transformer into a probabilistic one to handle uncertainty. The system also incorporates a posterior reasoner to refine and validate information. UPD-Trans is evaluated on VAR dataset and exhibits improved reasoning performance.

**Strengths:**

1.	The proposed approach is innovative, demonstrating a clear advancement in the field.
2.	The methodology is articulately outlined and founded on technically robust principles, ensuring a strong basis for the research.
3.	The methodology includes thorough ablation studies that validate the effectiveness of each component of the proposed model.

**Limitations:**

1.	The rationale behind utilizing uncertainty estimation to aid models in visual abduction reasoning is well-founded. However, quantifying uncertainty remains a formidable challenge. Even when uncertainty is explicitly modeled through Gaussian random variables, the accuracy of the model’s predicted uncertainty can still be questionable. The experimental results depicted in Table 2 indicate that the enhancements provided by the uncertainty estimation module are limited.
2.	While the model achieves impressive performance on the VAR dataset, there is a risk of overfitting to this specific dataset’s characteristics. The generalizability of the model to other datasets or real-world scenarios remains to be thoroughly tested.
3.	The paper lacks comparisons with the most recent works published in 2023.
4.	The complexity of the proposed model, involving multiple encoders and decoders with cascaded reasoning, might lead to high computational costs, making it less practical for real-time applications.

**Suitability:**

2

---

### Meta-Review · Program_Chairs · 2024-07-13

**Recommendation:** Accept (Poster)
**Confidence:** 4

**Metareview:**

This paper introduces a probabilistic model named Uncertainty-Guided Probabilistic Distillation Transformer (UPD-Trans), which helps in explaining incomplete observations of the real world. Reviewers tend to align in the acceptance of the manuscript due to the novelty of the approach and the methodology (in particular, the ablation study). Still, there are concerns related to the benchmark (not including recent advances in 2023 for comparison purposes and its suitability to the conference with very few references to multimedia venues). This is a borderline paper, that can be presented as a poster.